# Gene Expression Profiling Elucidates Cellular Responses to NCX4040 in Human Ovarian Tumor Cells: Implications in the Mechanisms of Action of NCX4040

**DOI:** 10.3390/cancers15010285

**Published:** 2022-12-31

**Authors:** Birandra K. Sinha, Erik J. Tokar, Jianying Li, Pierre R. Bushel

**Affiliations:** 1Laboratory of Mechanistic Toxicology, Division of Translational Toxicology, National Institute of Environmental Health Sciences, Research Triangle Park, Durham, NC 27709, USA; 2Integrative Bioinformatics Group, National Institute of Environmental Health Sciences, Research Triangle Park, Durham, NC 27709, USA; 3Biostatistics and Computational Biology Branch, National Institute of Environmental Health Sciences, Research Triangle Park, Durham, NC 27709, USA

**Keywords:** microarray, gene expression, nitric oxide, TNFα, NCX4040, ROS

## Abstract

**Simple Summary:**

The nitric oxide donor, NCX4040, a non-steroidal anti-inflammatory-NO donor, is cytotoxic to a number of human tumors, including ovarian tumors cells. It has been shown to be safe in vivo. While the precise mechanism of action of NCX4040 is not clear at this time, NCX4040 has been reported to generate both reactive oxygen species (ROS) and reactive nitrogen species (RNS), causing significant DNA damage in tumor cells. In this study, we provide evidence that NCX4040 induces the differential induction of oxidative stress genes, inflammatory response genes (TNF, IL-1, IL-6 and COX2), DNA damage response and MAP kinase response genes in human ovarian tumor cells. Our studies strongly suggest that formation of ROS/RNS from both induction of NOX and CHAC1 which causes a significant depletion of GSH that results in oxidative stress. Furthermore, NCX4040 treatment induced TNF-dependent pathways which further increases oxidative stress in ovarian tumor cells, resulting in enhanced NCX4040-mediated cell death from apoptosis and/or ferroptosis.

**Abstract:**

The nitric oxide donor, NCX4040 is a non-steroidal anti-inflammatory-NO donor and has been shown to be extremely cytotoxic to a number of human tumors, including ovarian tumors cells. We have found that NCX4040 is cytotoxic against both OVCAR-8 and its adriamycin-selected OVCAR-8 variant (NCI/ADR-RES) tumor cell lines. While the mechanism of action of NCX4040 is not entirely clear, we as well as others have shown that NCX4040 generates reactive oxygen species (ROS) and induces DNA damage in tumor cells. Recently, we have reported that NCX4040 treatment resulted in a significant depletion of cellular glutathione, and formation of both reactive oxygen and nitrogen species (ROS/RNS), resulting in oxidative stress in these tumor cells. Furthermore, our results indicated that more ROS/RNS were generated in OVCAR-8 cells than in NCI/ADR-RES cells due to increased activities of superoxide dismutase (SOD), glutathione peroxidase and transferases expressed in NCI/ADR-RES cells. Further studies suggested that NCX4040-induced cell death may be mediated by peroxynitrite formed from NCX4040 in cells. In this study we used microarray analysis following NCX4040 treatment of both OVCAR-8 and its ADR-resistant variant to identify various molecular pathways involved in NCX4040-induced cell death. Here, we report that NCX4040 treatment resulted in the differential induction of oxidative stress genes, inflammatory response genes (*TNF*, *IL-1*, *IL-6* and *COX2*), DNA damage response and MAP kinase response genes. A mechanism of tumor cell death is proposed based on our findings where oxidative stress is induced by NCX4040 from simultaneous induction of *NOX4*, *TNF-α* and *CHAC1* in tumor cell death.

## 1. Introduction

Ovarian cancer affects a large number of women and is a leading cause of death among gynecological cancers in most developed countries. Furthermore, like breast cancer, ovarian cancers are highly heterogeneous and consist of serous carcinomas (SC), mucinous carcinomas (MC), endometrioid carcinomas (EC), and clear-cell carcinomas (CCC), transitional-cell Brenner tumors, mixed, and undifferentiated type [1]. While these subtypes have distinct etiology, morphology, molecular biology and prognosis, these are treated as a single cancer. The standard treatment consists of cytoreductive surgery and combination chemotherapy with cis-platin, taxanes and/or adriamycin [2]. While the response rate to first-line therapy is reported to be around 80–90%, most patients relapse and develop chemotherapy resistance. Taxanes, as well as adriamycin, are known substrates of P-gp, an ATP-dependent transporter protein highly expressed in various clinical tumor samples. Because of the emergence of drug resistance and failure to chemotherapy of ovarian cancers, there is an urgent need for better clinically active drugs. Our previous studies have shown that NCX4040, a non-steroidal nitric oxide donor, is active against both OVCAR-8 and its adriamycin-resistant OVCAR-8 (NCI/ADR-RES) ovarian tumor cells [3,4]. NCX4040 is a nitro derivative of aspirin and has been reported to generate ^•^NO following cellular activation [5,6]. We have shown that NCX4040 depletes glutathione and generates ROS, leading to oxidative stress, DNA damage and tumor cell death [4]. Most interestingly, we have found that NCX4040 also inhibits functions of ATPase of the transporter proteins [3]. We have shown that ^•^NO generated from various NO-donors is extremely effective in inhibiting ATPase activities of ATP-dependent ABC transporters which leads to reversal of drug resistance in human MDR tumor cells [7,8]. We have found that NCX4040, at subtoxic concentrations, was very effective in reversing drug resistance in both P-gp- and BCRP-overexpressing human cancer cell lines [3].

While the precise mechanism of tumor cell death by NCX4040 is not clear, NCX4040 treatment causes depletion of cellular glutathione, generates reactive oxygen species (ROS), and induces DNA damage [5,9]. Recently, we have shown that in addition to ROS, NCX4040 also generated reactive nitrogen species (RNS) which leads to cellular death in ovarian cells [3,4]. Because NCX4040 inhibited ATPase activities of ABC transporters and was active against adriamycin-resistant ovarian tumor cells, we sought to further examine the mechanism of action of NCX4040 in detail as we believe that by understanding the mechanism of action of NCX4040, better and more active anticancer agents can be designed which may be highly effective for the treatment of human cancers in the clinic. Therefore, we have utilized global gene microarray analysis in ovarian tumor cells to further decipher mechanism of NCX4040-dependent cell death. Our results show that NCX4040 treatment of ovarian cells induced genes involved in DNA damage response and repair. Furthermore, NCX4040 induced both the TNF-dependent pathway and *CHAC1*, leading to both GSH depletion and oxidative stress.

## 2. Materials and Methods

### 2.1. Materials

NCX4040 was purchased from Sigma Chemicals (St. Louis, MO, USA) and was dissolved in DMSO. Stock solutions were stored at −80 °C. Fresh drug solutions, prepared from the stock solutions, were used in all experiments.

### 2.2. Cell Culture and Drug Treatment

Authenticated human OVCAR-8 and its adriamycin-resistant National Cancer Institute (NCI/ADR-RES) tumor cells were obtained from NCI, Frederick, MD. Cells were grown in Phenol Red-free RPMI 1640 media supplemented with 10% fetal bovine serum and antibiotics. Tumor cells were routinely used for 20–25 passages, after which the cells were discarded, and a new cell culture was started from the frozen stock. Three independent experiments of exponentially growing cells (65–70% confluency) were left untreated or treated with NCX4040 (5 µM) for 4, 24 and 48 h and then washed twice with ice-cold PBS (pH 7.4). Total RNA was extracted with TRIzol (Ambion, Life Technologies, Grand Island, NY, USA) and RNeasy mini kit columns (Qiagen, Valencia, CA, USA).

### 2.3. Global Gene Expression and Data Acquisition

Gene expression analysis was conducted using Agilent Whole Human Genome 4 × 44 multiplex format oligo arrays (014850) (Agilent Technologies, Santa Clara, CA, USA) following the Agilent 1-color microarray-based gene expression analysis protocol. Starting with 500 ng of total RNA, Cy3 labeled cRNA was produced per manufacturer’s protocol. For each sample, 1.65 µg of Cy3 labeled cRNAs were fragmented and hybridized for 17 h in a rotating hybridization oven. Slides were washed and then scanned with an Agilent Scanner. Data was obtained using the Agilent Feature Extraction software (v12), using the 1-color defaults for all parameters. The Agilent Feature Extraction Software performed error modeling, adjusting for additive and multiplicative noise. The data is available in the Gene Expression Omnibus under GEO accession GSE212017.

### 2.4. Preprocessing of the Data

Pixel intensity values of probes that map to the same gene were averaged. The data was log_2_ transformed and quantile normalized.

### 2.5. Statistical Data Analysis

The following three-way analysis of variance (ANOVA) was used to model the log_2_ quantile normalized data:*Y_ijkl_* = *μ* + C_i_ + T_j_ + D_j_ + (C × T × D)_ij_+ *ε_ijkl_*
where *μ* is grand mean of the experiment, *Y_ijkl_* represents the *k*th gene expression observation on the *i*th cell line (C), *j*th treatment (T), *k*th duration and *ε_ijkl_* the random error assumed to be normally and independently distributed with mean 0 and standard deviation *δ* for all measurements. Fisher’s least significant difference *t*-test was performed for each gene to compare the mean of the treated samples and WT- (or R-) to the mean of the control- samples. There were a total of 1851 differentially expressed genes (DEGs) detected at a Benjamini and Hochberg [10] false discovery rate (FDR) < 0.05 and absolute fold change > 2.0 (Appendix A). Using the Database for Annotation, Visualization, and Integrated Discovery (DAVID) v6.8 [11,12] the 1851 DEGs were enriched for KEGG pathways at an FDR ≤ 0.05 with minimum category size ≥ 5.

### 2.6. Pattern-Driven Analysis of Gene Expression

To identify co-expression patterns in the 1851 DEGs, ratio values for each gene were generated by subtracting the average log_2_ quantile normalized pixel intensity of the time-matched controls from the log_2_ quantile normalized pixel intensity of the time-matched samples. The ratio values were then analyzed to extract patterns and identify co-expressed genes (EPIG) using the EPIG software [13] with the parameters Pearson correlation ≥ 0.7, signal/noise ≤ 2.5, minimum pattern size ≥ 6, magnitude of log_2_ fold change > 0.5 and *p*-value < 0.0001.

### 2.7. Real-Time RT-PCR

The expression levels of selected transcripts were confirmed by real-time polymerase chain reaction (RT-PCR) using absolute SYBR green ROX Mix (ThermoFisher Scientific, Rochester, NY, USA) as previously described [14]. Data were analyzed using ΔΔCt method of relative quantification and to determine transcript levels, cycle times (Ct) were compared from the same sample and normalized to β-actin (or GADPH) of untreated controls, which were set at 100%. Primers for the selected genes were designed using Primer Express 1.0 software and in some cases were synthesized (Integrated DNA technologies, San Diego, CA, USA) from published literature or were purchased from Origene (Gaithersburg, MD, USA). All real-time fluorescence detection was carried out on an iCycler (Bio-Rad, Hercules, CA, USA). All experiments were carried out three different times and results are expressed as the mean ± SEM. Analysis were performed using unpaired Student’s *t*-test and considered significant when *p* ≤ 0.05.

## 3. Results

### 3.1. Analysis of Overlay of DEGs by Vaan diagrams

To identify DEGs from samples exposed to 5 µM of NCX4040 for 4, 24 and 48 h, we analyzed microarray gene expression data with a three-way analysis of variance (ANOVA) model that incorporated OVCAR-8 (WT) and NCI/ADR-RES (R) cells as a factor, duration of exposure as a factor, treatment of the compound and the interaction of the three as a factor. The ANOVA was followed by Fisher’s least significant difference *t*-tests to compare the mean of treated samples to the mean of control samples (0 time point). The analysis revealed 1851 DEGs based on an absolute fold change > 2.0 and FDR < 0.05. Venn diagrams of the overlap of the DEGs are shown in Figure 1. It is interesting to note that treatment with NCX4040 at 4 h and 48 h resulted in a significant number of genes that were either upregulated or downregulated in both cell lines. In addition, Venn diagrams also show that a significant number of genes, common to both cells, were co-expressed following the drug treatment at 48 h when cells start to show NCX4040 cytotoxicity. This would suggest a common pathway for cell killing by NCX4040 in both OVCAR-8 and NCI/ADR-RES cells.

### 3.2. NCX4040 Induces Oxidative Stress-Related Genes

Analysis of the microarray data also indicated that a number of genes related to oxidative stress were differentially modulated by NCX4040 in both cell lines at 4 and 48 h treatment (Table 1). Results show that *HMOX/OX*, a marker for oxidative stress [15,16], was significantly (6.7-fold and 11.1-fold) elevated in both OVCAR-8 and NCI/ADR-RES cells, respectively at 4 h following NCX4040 treatment. Furthermore, *SOD2,* an enzyme responsible for removing superoxide anion radical [17], was induced more than 2-fold by NCX4040 in OVCAR-8 cells at both 4 and 48 h of treatment. *NOX4*, an NADPH oxidase, which has been suggested to be a major generator of O_2_^•−^ and H_2_O_2_ by transferring electrons from NADPH to O_2_ [18], was significantly induced following NCX4040 treatment at 4 h in both cell lines and remained elevated at 48 h in OVCAR-8 cells. *CHAC1* (glutathione-specific γ-glutamylcyclotransferase 1) which catalyzes the cleavage of glutathione into 5-oxo-L-proline and a Cys-Gly dipeptide was also significantly induced by NCX4040 treatment in both cells. Induction of *CHAC1* gene has been shown to lead to a significant depletion of cellular glutathione, resulting in ROS formation and increase in oxidative stress [19].

### 3.3. NCX4040 Induces Inflammatory Response Genes

Microarray analysis also indicated that various inflammatory response genes were significantly modulated by NCX4040 treatment in ovarian cells (Table 2). It is interesting to note that IL-6 gene, a pleiotropic cytokine involved in inflammation, hematopoiesis, bone metabolism, and embryonic development was significantly induced at 4 h and remained elevated at 48 h in both cells [20,21]. Other inflammation response genes (e.g., *NF-kB*, *TNF*, *IL-1*, Table 2), [22] were also significantly induced by NCX4040 in both cells suggesting that NCX4040 induces inflammation by inducing ROS formation and causing oxidative stress.

### 3.4. NCX4040 Modulates DNA Damage Response Genes

As previously reported, NCX4040 induces DNA double-strand breaks [4,5]. Our microarray analysis indicated that various DNA damage response genes were also induced (or decreased) following NCX4040 treatment (Table 3). We found that both *BRCA1* and *BRCA2* were downregulated in both cells. *BRCA1* and 2 have been reported to be involved in repairing double-strand DNA breaks via the homologous recombination repair (HRR) pathway [23,24]. Various *XRCC* genes were found to be downregulated by NCX4040. *XRCCs* are proteins that are involved in both base excision and the single-strand break *repair* [25], indicating decreased repair activities of DNA damage induced by NCX4040. Microarray analysis also indicated that NCX4040 modulated both *RAD51* and *GADD45* genes. *RAD51* has been shown to be specific for repairing a DNA double-strand break in cells. *GADD45*, a growth arrest and DNA damage response gene, was also found to be significantly up-regulated in NCI/ADR-RES cells.

### 3.5. Principal Component Analysis

Principal component analysis (PCA) of the 1851 DEGs is shown in Figure 2. The 1st three principal components (PC) capture 73% of the variability in the data. As can be seen, the biological replicates are projected in 3D space very close to each other. In addition, PC2 separates the samples by duration of exposure whereas PC3 separates the samples by cell line.

### 3.6. Analysis of Clustering of DEGs (Heatmap)

Figure 3 shows the clustering of the 1851 DEG’s and a heatmap of the differential expression of the genes following treatment with NCX4040 of both cells at 4, 24 and 48 h. The patterns of expression reveal a time dependency in the NCX4040 exposure that differs between the WT and the resistant cells. In both cell lines, the 48 h exposure elicited the largest number of genes that are up- or down-regulated by NCX4040 treatment.

### 3.7. KEGG Pathway Analysis

Enrichment analysis of the 1851 DEG’s using KEGG pathways following NCX4040 treatment at 4, and 48 h is shown in Table 4. It is noteworthy that at 24 h treatment, no significant differences were observed (data not shown) (Appendix A). Our results show that a number of important pathways were differentially affected by NCX4040 in these two cell lines. In the WT OVCAR-8 cells, NCX4040 significantly induced pathways related to TNF and cell cycle while in the resistant cells, NCX4040 modulated pathways related to both MAP kinase and DNA replication.

### 3.8. Analysis for Co-Expression Pattern

To uncover co-expressed genes that are regulated coordinately from NCX4040 exposure in a time dependent manner, we analyzed the 1851 DEGs using the Extracting Patterns and Identifying co-Expressed genes (EPIG) software with expression profiles correlated > 0.7, having signal to noise ≤ 2.5, a magnitude of fold change > 0.5 and within patterns of size ≥ 6. EPIG extracted 1555 gene expression profiles (Appendix A) into 17 patterns with a *p*-value of significance < 0.0001 (Appendix A, Figure 4 and Figure 5). Of note are genes in pattern #6 that are upregulated early at the 4 h exposure in both cell lines. Some of the 121 genes enrich for the MAPK signaling pathway. On the other hand, genes in patterns #12 and #13 are down-regulated early at the 4 h exposure in both cell lines. Conversely, gene in pattern #17 are downregulated in both cell lines and relatively unchanged at the 4 h exposure. Pattern #14 contain late response genes that are down-regulated at the 48 h exposure mark in both cell lines and pattern #11 consists of genes that are upregulated in the WT cells but down-regulated in the resistant cells. A significant number of 116 genes enrich for regulation of cytokine production. Interestingly, there are genes in patterns #2 and #3 that cycle in both cell lines. They are up at 4 h, down at 24 h and back up at 48 h. A significant number of the 58 genes enrich for the TNF signaling pathway.

### 3.9. RT-PCR Analysis

Our microarray analysis indicated that various genes representing oxidative stress, DNA response and inflammatory response pathways (Table 1, Table 2 and Table 3) were modulated by NCX4040 treatment. We sought to confirm this by RT-PCR analysis in these ovarian cells and are presented in Table 5. It can be seen that there is a significant agreement between microarray and RT-PCR analysis.

### 3.10. Cell Cycle Pathway Analysis by IPA

An IPA molecular interaction network for the cell cycle pathway overlayed with DEGs induced by NCX4040 in a time-dependent manner is shown in Figure 6. GA45A and CDN1A have similar expression profiles across the time points in both cell lines except for the expression in the former is upregulated more at 48 h in the resistant cells. Conversely, SCF and CCNE2 have similar expression across the time points in both cell lines except for the expression in the former is more upregulated in the wild type cells.

### 3.11. NCX4040 Induces TNF Signaling Pathway

Microarray analysis also identified pathways related to TNFα and MAPK (Table 4) in these ovarian tumor cells following NCX4040 treatment. Inflammation response genes are shown in Table 2 and some these genes were confirmed by RT-PCR (Table 5). TNFα is a pleotropic cytokine that is known to initiates many downstream signaling pathways, including NF-kB activation, MAP kinase activation, and has been reported to induce cell death via apoptosis and necrosis [26,27,28]. TNFα has shown to lead to ROS generation through activation of NADPH oxidase, and a close relationship has been found to exist between ROS/RNS and TNF [29]. TNF has been suggested to be a regulator for the formation of both ROS and RNS in cells, leading to cell death [29,30,31]. A representative TNF-signaling pathway leading to generation of ROS, cell death and induction of inflammation by NCX4040 is shown in Figure 7.

## 4. Discussion

DNA microarray analysis is used frequently to identify characteristics of tumors as well as molecular mechanisms of drugs in complex cellular systems [32,33,34,35,36,37]. Previously, we have utilized global gene microarray analysis to identify molecular pathways involved in topotecan-induced cell death in human breast tumor cells [14]. To further elucidate the molecular pathways responsible for cell death in these ovarian tumor cell lines and to design selective inhibitors of ATP-transporters, we have now carried out global gene microarray analysis following treatment with NCX4040. Since NCX4040 generates ROS in tumor cells we chose these ovarian cancer cell lines (OVCAR-8 and NCI/ADR-RES) to decipher the mechanisms of NCX4040 as NCI/ADR-RES cells show resistance to free radical generating drugs, including adriamycin due to higher expressions of SOD, catalase, glutathione-dependent peroxidase and transferase in NCI/ADR-RES cells [38,39]. Functions of these enzyme/proteins is to protect cells from harmful damage (and cell death) by eliminating ROS and other reactive species from cells.

Our microarray analysis has identified four major pathways induced by NCX4040 in these ovarian cells (Table 4). Of significant interests are TNF-signaling and MAPK pathways. Furthermore, we have found that various genes involved in oxidative stress pathways, DNA damage response and inflammation response genes were differentially expressed by NCX4040 treatment in both cell lines. We used RT-PCR analysis to confirm representative genes belonging to DNA damage response, inflammation- and oxidative stress response. While there is a significant agreement between microarray and RT-PCR analysis, there are also some differences in fold increase/decrease in gene expressions, e.g., *IL-6*. These may be due to differences in methods as well as primers used for detecting these genes.

Our recent studies, as well as that of others, have shown that NCX4040 induces oxidative stress and generates ROS [4,5]. This is further confirmed in this study by significant inductions of genes involved in protection of ROS-mediated cell death in these ovarian cells. We found that heme oxygenase 1 (*HMOX1)* was significantly induced in both cell lines. HMOX1 is involved in protection of cellular death by ROS. Furthermore, *SOD2*, a gene responsible for removing cellular superoxide anion radical before it’s dismutation to reactive peroxide and iron-catalyzed formation of ^•^OH, was significantly induced in OVCAR-8 cells (Table 1). Superoxide anion radical appears to be generated in these cells as *NOX4* was also induced by NCX4040 in both cells. *NOX4* is a NADPH oxidase and is responsible for catalyzing intracellular conversion of O_2_ into O_2_^•−^ and other ROS. Most interestingly, microarray analysis indicated that *CHAC1* was significantly induced in both cell lines at 4 h and 48 h of treatment with NCX4040. CHAC1 gene encodes a member of the gamma-glutamylcyclotransferase1 family of proteins which are important in regulation of glutathione levels and elevated levels in *CHAC1* has been shown to result in GSH depletions and increase in oxidative stress in cells [19]. Furthermore, it has been suggested that *CHAC1* is also involved in unfolded protein response, resulting from oxidation of essential -SH functions of proteins and ER stress [40]. The encoded protein has been shown to promote neuronal differentiation by deglycination of the Notch receptor, which prevents receptor maturation and inhibits Notch signaling pathways. Chen et al. have shown that overexpression of *CHAC1* results in inhibition of Notch-3-mediated down-stream signaling [41]. In addition, it induced cell death by activating caspase3/9, increased ROS formation and caused modulation of temozolomide cytotoxicity against glioma tumor cells [41]. We have recently reported that peroxynitrite, formed from O_2_^•−^ and ^•^NO generated from NCX4040 in cells, may be involved in cell death of ovarian tumor cells [4]. We found that significant amounts of DNA damage was introduced in OVCAR-8 cells; however, decreased amounts of DNA damage were observed in the resistant NCI/ADR-RES cells, resulting in decreased cell death. While the nature of cell death was not studied in our earlier study [4], recent studies suggest that ferroptosis may be caused by ^•^NO and ROS generation in cells [42]. Furthermore, it is very interesting to note that induction of *CHAC1* and *NOX4*, two genes significantly induced by NCX4040, have been suggested to be biomarkers for ferroptosis-related cell death in tumor cells [43,44]. Further studies with NCX4040 are in progress to confirm this in ovarian and other tumor cells.

Important findings of our work described here are that NCX4040 induces a significant amount of ROS/RNS generation by inducing NOX4 enzymes which are responsible for the generation of O_2_^•−^ in tumor cells, and induction of *CHAC1* which leads to depletion of cellular GSH and oxidative stress. Furthermore, our results show that NCX4040 induces TNF-dependent signaling pathways, which further increase ROS formation, cellular oxidative stress and cell death. These events, taken together, suggest that NCX4040 generates excessive amounts of ROS/RNS which are not removed from cells, leading to damage to various cellular components, including DNA and lipids. Our previous study [4], as well as that of others [5], have indicated NCX4040 induces cellular DNA damage. Our current study shows that various DNA damage response genes are modulated by NCX4040, and that DNA damage is only partially repaired in cells which may lead to cell cycle arrest as reported here (Figure 6). High ROS/RNS formation in cells may also lead to oxidation of cellular molecules, including proteins that are involved in folding (ER stress) as it has been shown that oxidative stress causes ER stress [45,46,47]. It is also possible that high ROS/RNS may also oxidize various lipids, generating toxic intermediates which may also cause more damage to cellular components as suggested during ferroptosis and cell death [48,49,50]. A proposed mechanism of cell death induced by NCX4040 is shown in Figure 8.

## 5. Conclusions

In conclusion, our studies show that NCX4040 treatment of ovarian tumor cells results in the differential induction of oxidative stress genes, inflammatory response genes (*TNF*, *IL-1*, *IL-6* and *COX2*), DNA damage response and MAP kinase response genes. Our studies strongly suggest that formation of ROS/RNS from both induction of *NOX* and *CHAC1* causing a significant depletion of GSH that results in oxidative stress. Furthermore, NCX4040 treatment also induced TNF-dependent pathways which further increases oxidative stress in ovarian tumor cells, resulting in enhanced NCX4040-mediated cell death from apoptosis and/or ferroptosis.

## Figures and Tables

**Figure 1 cancers-15-00285-f001:**
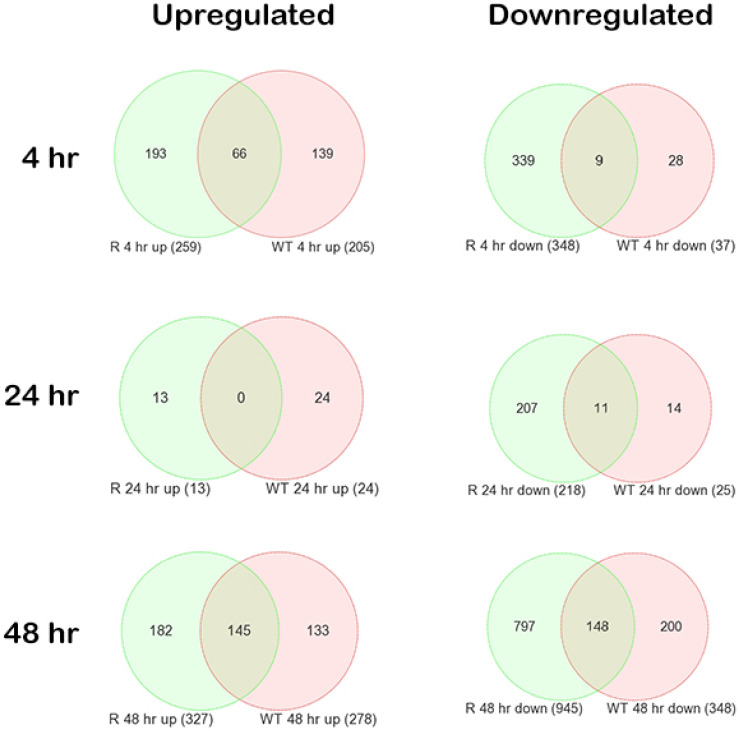
Venn diagrams of the overlap of the differentially expressed genes (DEGs) (treated to 0 time point) based on an absolute fold change > 2.0 and FDR < 0.05. WT is the OVCAR-8 wild type and R is the adriamycin-resistant (NCI/ADR-RES) cells. The rows represent the time of exposure (in hours) to 5 µM of NCX4040. The Columns represent the directionality of the DEGs.

**Figure 2 cancers-15-00285-f002:**
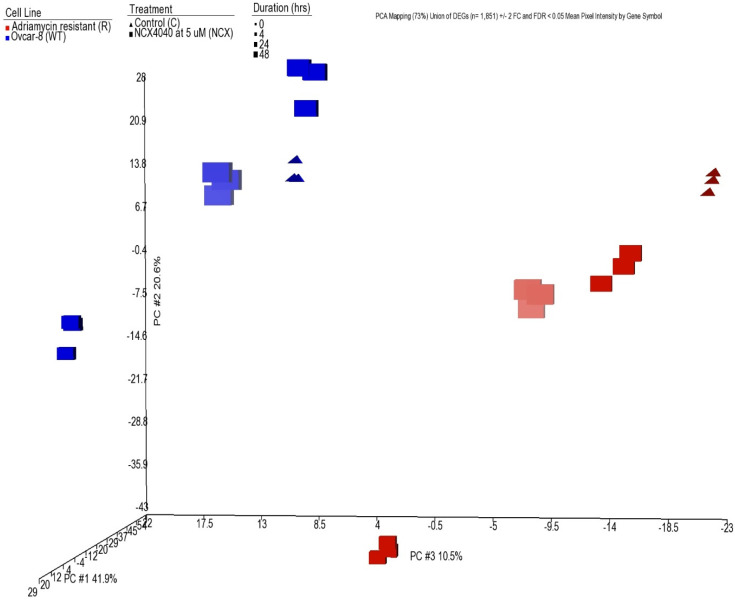
Principal component analysis of the samples. Principal component analysis of the union of the DEGs (*n* = 1851) from Figure 1 was performed to project the samples in 3-dimensional space based on the 1st three principal components (PC #1, PC #2 and PC #3). The mean log base 2-pixel intensity data for each gene was used. The numbers following the PCs represent the amount of variability captured by the components. The total variability captured by the three PCs is 73%. PC #3 is the x-axis; PC #2 is the y-axis and PC #1 is the z-axis. Cell line is represented by color: blue for ovcar-8 (WT) and red for adriamycin resistant (R). Treatment is represented by shape: Triangle for the control (C) and square for 5 µM NCX4040 (NCX). Duration is represented by the size: smallest for 0 h to largest for 48 h. The lighter shading of a point denotes projection towards the back of the plot.

**Figure 3 cancers-15-00285-f003:**
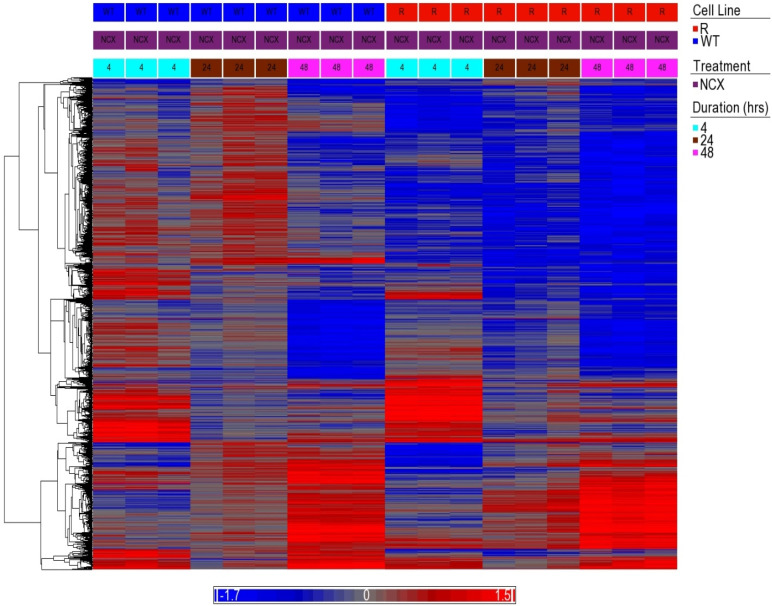
Hierarchical clustering heat map of DEGs. Average linkage for grouping and cosine correlation as a dissimilarity measure were used to hierarchical cluster the union of the DEGs (*n* = 1851) from Figure 1 based on the log base 2-pixel intensity ratio (treated to control) data. The genes are in the rows and the samples are in the columns. The color legend depicts the log base 2-pixel intensity ratio values of the gene in the sample. Red is upregulated, blue is downregulated, and grey is no change. WT is the OVCAR-8 wild type and R is the adriamycin-resistant (NCI/ADR-RES) cells. Treatment with 5 µM NCX4040 for 4 h (aqua), 24 h (brown) or 48 h (pink).

**Figure 4 cancers-15-00285-f004:**
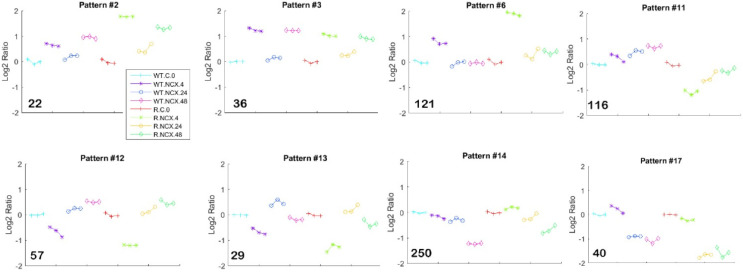
Co-expression patterns. The Extracting Patterns and Identifying co-Expressed Genes (EPIG) method was used to identify gene sets with co-expression patterns across the samples. Depicted are 8 of the 17 patterns containing 1555 genes expression profiles with significant (*p*-value < 1 × 10^−4^) co-expression > 0.7 Pearson correlation and magnitude of fold change > 0.5. The number in the lower left corner of each panel represents the number of genes in the pattern. The patterns are representative of log base 2 ratio value of the top 6 correlated gene expression profiles in the pattern. WT is the ovcar-8 wild type and R is the adriamycin-resistant variant. Treatment with 5 µM NCX4040. Time of exposure is 0, 4, 24 or 48 h. The x-axis is the samples represented by the line colors and data point shapes in the legend. The y-axis is the log base 2 ratio value.

**Figure 5 cancers-15-00285-f005:**
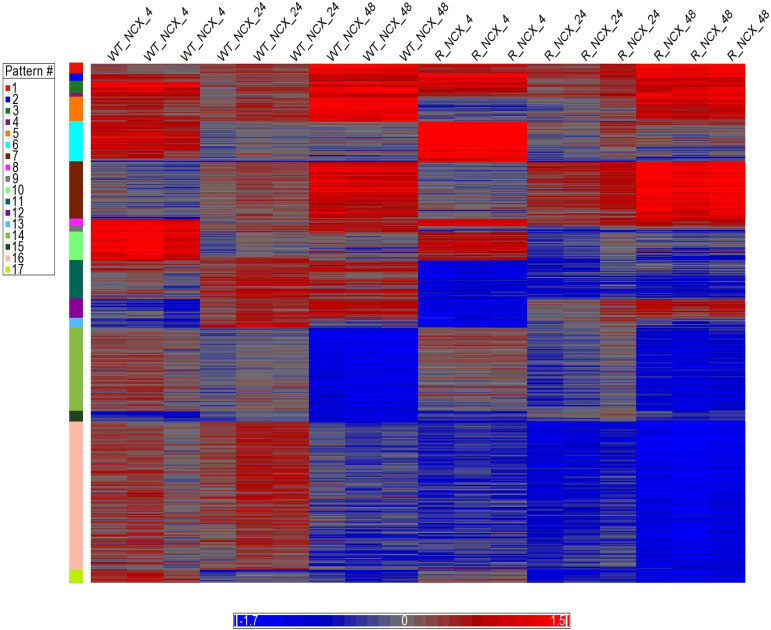
Hierarchical clustering heat map of EPIG pattern genes. Average linkage for grouping and cosine correlation as a dissimilarity measure were used for hierarchical clustering of the EPIG pattern genes (*n* = 1555) from Figure 4 based on the log base 2-pixel intensity ratio data. The co-expressed gene sets in the patterns are in the rows and the samples are in the columns. The pattern # legend shows the colors of the 17 patterns. The color legend depicts the log base 2-pixel intensity ratio values of the gene in the sample. Red is upregulated, blue is downregulated, and grey is no change. WT is the OVCAR-8 wild type and R is the adriamycin-resistant (NCI/ADR-RES) cells. Cells were treated with 5 µM NCX4040 for 4, 24 or 48 h.

**Figure 6 cancers-15-00285-f006:**
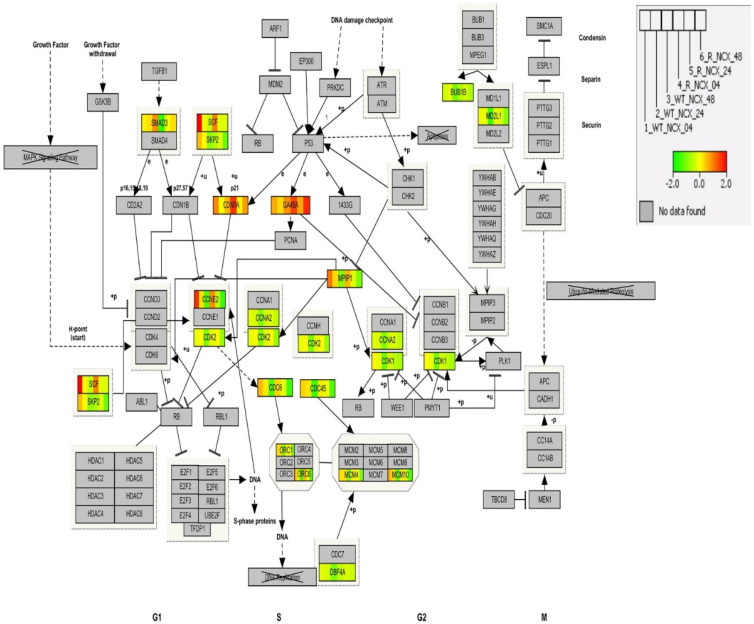
Overlay of gene expression on the cell cycle pathway. The human cell cycle signaling pathway is overlaid with the average of the log base 2-pixel intensity ratio values from the replicate samples. Red denotes induction, green repression and yellow no change. Gray indicates that the gene was not mapped. The legend illustrates the segmentation of the genes according to the data from a given treatment. WT is the OVCAR-8 wild type and R is the adriamycin-resistant (NCI/ADR-RES) cells. Cells were treated with 5 µM NCX4040 for 4, 24 or 48 h.

**Figure 7 cancers-15-00285-f007:**
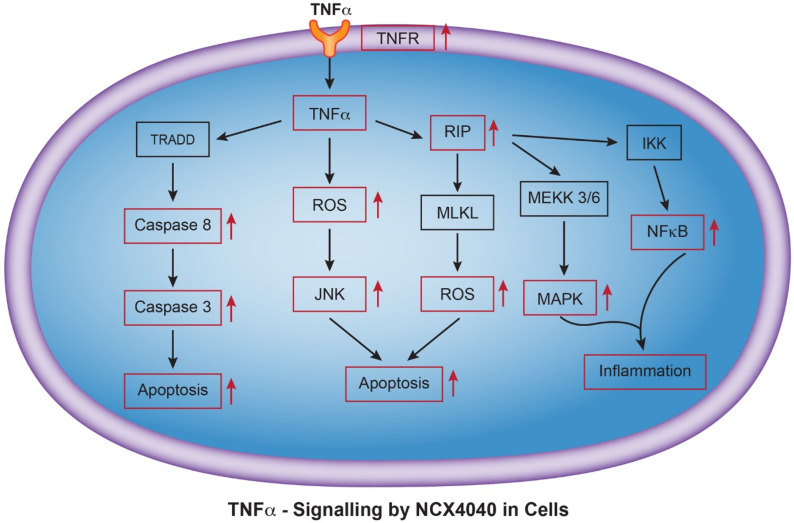
TNFα signaling, inflammation and cell death induced by NCX4040 in OVCAR-8 and NCI/ADR-RES cells. Abbreviation used: TRADD, tumor necrosis factor receptor type-1 associated death domain; RIP, receptor interacting protein; JNF, c-jun N-terminal kinase, IKK, IkappaBkinase; MEKK or MAPK, mitogen activated protein kinase; MLKL, mixed lineage kinase domain. Red boxes represent genes/pathways induced by NCX4040 in ovarian tumor cells.

**Figure 8 cancers-15-00285-f008:**
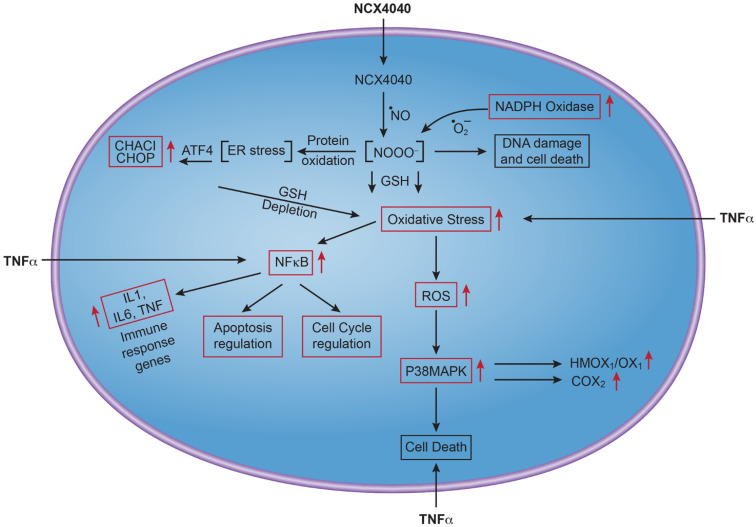
Proposed pathways for tumor cell death induced by NCX4040. Of note is formation of ROS/RNS from both induction of *NOX*, and *CHAC1*, resulting in significant GSH depletion and increased oxidative stress, TNF formation, immune response-mediated by NF-kB induction and finally NCX4040-mediated apoptosis (or ferroptosis). Red boxes/arrows represent genes induced by NCX4040 in ovarian tumor cells.

**Table 1 cancers-15-00285-t001:** Oxidative stress genes differentially regulated following NCX4040 (5 µM) treatment.

Time	Gene	OVCAR-8	NCI/ADR-RES
4 h	HMOX1/OX1	+6.7	+11.1
	SOD2	+2.2	ND
	NOX4	+2.2	+2.1
	CHAC1	ND	+3.5
48 h	SOD2	+2.8	ND
	CHAC1	+3.8	+6.0

Data were obtained by microarray analysis and are expressed as fold change from controls. ND, not detected.

**Table 2 cancers-15-00285-t002:** Inflammatory response genes differentially expressed following NCX4040 (5 µM) treatment.

Time	Gene	OVCAR-8	NCI/ADR-RES
4 h	IL-6	+6.1	+2.2
	HSP	+3.5	+8.2
	MT1M	+3.4	+2.8
	MT1X	+2.1	+2.6
	TNFR	+3.3	+6.0
	TGF	+2.5	ND
	NFKB1	+2.2	ND
	IL1R	+2.1	ND
48 h	IL-6	+4.5	+4.2
	TNFR	+2.9	+2.2
	NFKB1	+3.9	+2.2
	VEGF	+3.5	+3.5
	IL-1β	+3.4	ND

Data were obtained by microarray analysis and are expressed as fold change from controls. ND, not detected.

**Table 3 cancers-15-00285-t003:** DNA response genes differentially expressed following NCX4040 (5 µM) treatment.

Gene	OVCAR-8	NCI/ADR-RES
BRCA1	−2.2	ND
BRCA2	−2.2	−2.0
RAD51	−3.2	−2.8
XRCC2	−2.3	ND
XRCC4	−2.3	ND
XRCC5	ND	−2.1
XRCC6	ND	−2.1
GADD45	ND	+3.2

Data were obtained by microarray analysis and are expressed as fold change from controls. These genes were not significantly changed at 4 h or 24 h. ND, not detected.

**Table 4 cancers-15-00285-t004:** KEGG Pathway analysis of the 1851 DEGs following treatment with 5 µM NCX4040.

Cell Line and Treatment		KEGG Pathway	Count	%	*p*-Value	FDR
WT 4 h	TNF signaling pathway	10	7		1.10 × 10^−6^	7.50 × 10^−5^
WT 48 h	Cell cycle	14	3.3		5.40 × 10^−7^	3.80 × 10^−5^
R 4 h	MAPK signaling pathway	14	5.7		4.90 × 10^−53^	7.00 × 10^−2^
R 48 h	DNA replication	7	1.4		8.10 × 10^−5^	7.50 × 10^−3^

The number under KEGG Pathway is the number of genes that are annotated with the corresponding pathway term. Count is the percentage of DEGs in the corresponding pathway term.

**Table 5 cancers-15-00285-t005:** Analysis of representative genes differentially expressed by NCX4040 (5 µM) treatment by RT-PCR.

4 h	Gene	OVCAR-8	NCI/ADR-RES
	HMOX1/OX1	+4.5 ***	+7.3 ***
	CHAC1	1.0	1.0
	NFKB1	1.0	+5.2 ***
	VEGFA	1.0	+4.9
	IL-6	+4.6 ***	+16.0 ***
	COX2	1.0	+5.8 ***
	RAD51	+2.8 ***	0.8
	GADD45	+1.2	1.0
48 h	HMOX1/OX1	0.9	0.5 ***
	CHAC1	+8.0 ***	+3.6 ***
	NFKB1	1.0	+2.3
	VEGFA	+3.0 ***	+10.0 ***
	IL-6	+3.5 ***	+20.0 ***
	COX2	+2.0 *	+10.9 ***
	RAD51	0.4 **	0.23 **
	GADD45	+3.0 ***	0.5 *

Data are expressed as fold change from β-actin (or GADPH) used as the control (untreated cells at 0, 4 and 48 h). ***, **, * *p* values < 0.001, 0.005 and 0.05, respectively.

## Data Availability

Gene Expression Data and Appendix A are available on Gene Expression Omnibus under GEO accession GSE212017.

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
