# Peer review of "Gene Expression Profiling Elucidates Cellular Responses to NCX4040 in Human Ovarian Tumor Cells: Implications in the Mechanisms of Action of NCX4040"

_cancers, 2022, doi:10.3390/cancers15010285_

Round 1
Reviewer 1 Report (Previous Reviewer 3)
The authors have addressed all the issues in the manuscript.
Introduction has been improved, however with the new first paragraph introduced there is information repeated in the second paragraph that has not been revised accordingly. This should be corrected.
Subheadings have been introduced in the results session which makes it easier to follow.
Supplemental Table2 has been included.
Author Response
All corrections have been made. Introduction section has been corrected.
Reviewer 2 Report (Previous Reviewer 4)
The response to the comments looks promising. Thank you.
Author Response
All corrections have been made. Introduction section has been corrected.
Reviewer 3 Report (New Reviewer)
Gene Expression Profiling Elucidates Cellular Responses to NCX4040 in Human Ovarian Tumor Cells: Implications in the Mechanisms of Action of NCX4040 by Sinha et al. proposed a mechanism of tumor cell death induced by NCX4040.
This work can be interesting, is very little documented. All of the results proposed are based on current methodologies that can be used to screen whichever active ingredient. The fact that candidate molecule is possibly a pre-drug, to validate this hypothesis, animal experimentation is needed, in xenograft models.
So this work apart from the problems of writing, (the introduction for Ex.), it lacks the validation of the effectiveness of the product in vivo.
Therefore, this current status report is not ready for publication.
Author Response
Reviewer 3In vivo experiments were not carried out for two separate reasons:
First, we have a major problem at our Institute in doing any in vivo mouse experiment as everything has been backed up due to the shutdown of the Institute from COVID -19.
This has also resulted in decreased support personnel. Second, we have designed several analogs of NCX4040 based on our previous work and this study (structures of these analogs are attached). These are currently being synthesized. In- silico modelling
indicates these analogs bind better than NCX4040 to the ATP binding domain of P -gp and BCRP (Table for Binding Properties is also attached). Compound Number I I appears to be significantly better than NCX4040. Thus, it is anticipated that these analogs, especially compound II, may be better in inhibiting ATPase activities of ABC transporters and in reversing MDR. Once synthesized, we plan to evaluate these
analogs for their anti-cancer activities and choose the best analog to carry out in vivo studies along with NCX4040.
If we find that compound II (or other compounds) is significantly better than NCX4040 as an anticancer agent, we will ask our sister institute, NCI, to carry out anti -tumor activity determination in their 60- tumor cell line panel to identify other human tumors that are also sensitive to this compound.
Finally, I like to suggest to this reviewer please do not criticize other people’s writing when he/she cannot even write one complete sentence without a significant number of mistakes. Some of these sentences are neither complete nor make sense
Reviewer 4 Report (New Reviewer)
The manuscript titled “Gene Expression Profiling Elucidates Cellular Responses to NCX4040 in Human Ovarian Tumor Cells: Implications in the Mechanisms of Action of NCX4040” reports an in vitro study aimed at elucidating of the mechanism of the effect of NCX4040 on the gene expression profile of two different ovarian cancer cell lines, OVCAR-8 and its adriamycin-selected OVCAR-8 variant (NCI/ADR-RES). For this purpose, the authors utilized global gene microarray analysis in ovarian tumor cells to decipher the mechanism of NCX4040-dependent cell death. The authors found that NCX4040 treatment of ovarian cells induced genes involved in DNA damage response and repair. Furthermore, NCX4040 induced both the TNF-dependent pathway and CHAC1, leading to both GSH depletion and oxidative stress. In general, this is a well-designed and carried out regarding the gene expression analysis. However, a limitation of this study is that protein expression analysis was not performed. Therefore, the authors cannot be sure that those differentially expressed genes were accompanied by a corresponding increase/decrease in protein expression. Despite this limitation, this study reports interesting findings that contribute to understand the mechanism of action of NCX4040. The findings reported in this manuscript warrant further studies to characterize the protein expression profile induced by treatment of the same ovarian cancer cell lines with NCX4040. I found some minor issues that should be corrected in this manuscript.
Corrections
· The abbreviation SOC, used in the abstract, must be defined the first time it is used. It is also not included in the list of abbreviations.
· The text spanning lines 180-183 uses a larger font than that used in the rest of the manuscript. I don't see a reason for such a difference. Please, standardize the font size throughout the manuscript.
· In line 212 it is written XRCC’s to refers to the family of XRCCs protein. I think the correct form in XRCCs.
· Line 222: The of the 1,851 DEGs is shown in Figure-2. It seems that this sentence is incomplete.
· The authors state that there is a significant correlation between microarray and RT-PCR analysis and this result is shown in table 5. However, this table is a little confusing because the text explaining the meaning of each data shown is insufficient. Which analysis resulted in the p values shown in this table? I wonder if the correct wording would be "there is a significant agreement between microarray and RT-PCR analysis” instead of “there is a significant correlation...” For a better comparison, I suggest authors consider including the gene expression data obtained in the microarray analysis for the genes shown in this table.
Author Response
Our main research goal was to identify genes responsible for tumor cell death induced by NCX4040, which we have accomplished in this work. The reviewer is correct in that we do not know whether proteins will also be translated accordingly. We have initiated studies to identify roles of genes and related proteins, especially NOX4, GPx4 and CHAC1 in NCX4040-induced ferroptosis in various tumor cells by both pharmacological inhibitor/inducers and siRNA mediated gene silencing. This will be reported in due time.

This manuscript is a resubmission of an earlier submission. The following is a list of the peer review reports and author responses from that submission.
Round 1
Reviewer 1 Report
1. In this work, authors only used one wild-type ovarian cancer cell line OVCAR-8 and its mutant cell line which selected by adriamycin (NCI/ADR-RES) to perform the microarray. So it may lacke general significance.
2. The results haven’t confirmed via experiments neither in vitro nor in vivo.
Reviewer 2 Report
Major point
1. Why don't you describe that microarray-related data of NCX4040 treatment at 24h in both cell lines? Please, describe details microarray-related data of NCX4040 treatment at 24h in both cell lines.
2. I suggest that you need to carry out with statistical analysis. ex) Table 5.
Reviewer 3 Report
This study focuses on the compound NCX4040, a nitric oxide-releasing non-steroidal anti-inflammatory drug derived from aspirin, previously shown to be extremely cytotoxic to several human tumors, including ovarian tumors cells.
The authors have previously found that NCX4040 is cytotoxic against the ovarian cancer cell line OVCAR-8 and its adriamycin-selected multidrug-resistant variant (NCI/ADR-RES), leads to a significant depletion of cellular glutathione and to the formation of both reactive oxygen and nitrogen species (ROS/RNS), resulting in oxidative stress in these tumor cells.
In the present study the authors continued to work with the same ovarian cancer cell lines and used microarray analysis to identify the various molecular pathways involved in NCX4040-induced cell death.
This is an interesting study that contributes to expand the understanding of the mechanism of action of the novel compound NCX4040, with potential implications for its clinical use in cancer therapy.
The research design is appropriate for the goal of the work, although it is not clear why the authors used DNA microarrays to analyse gene expression instead of the RNA sequencing that allows for full sequencing of the whole transcriptome.
Major issues
1. Introduction must be improved to make clear the focus of the work. The first paragraph does not seem to be the most relevant information to start the introduction. On the other end, the first paragraph of Discussion seems more appropriate for the Introduction.
2. It is not clear what is the advantage of using DNA microarrays for gene expression analysis instead of RNA sequencing. Why were microarrays chosen for this study?
3. The Results section could benefit from Subheadings, it would be much easier to follow.
4. The presentation of the results must be improved. Description in the text does not always match with what is presented in the tables.
5. Could not find Supplemental Table2, file2.
6. In table 5 (validation of the microarray results by quantitative RT-PCR) it is not clear how the values were calculated. Are these relative to untreated cells? More explanations are needed regarding these results.
Minor issues
1. In tables, the title and caption should appear on top.
2. In line 67 appears NCX404 instead of NCX4040.
3. In line 184-185 gene names should be italicized.
4. In lines 196-199 – wrong text formatting.
5. In line 225, Figure 3 should not appear in bold.
6. Table 4 is not well formatted.
7. In Table 5. (4h) is mis localized in the heading.
8. In line 423 wrong formatting in (5. Conclusions).
Reviewer 4 Report
This manuscript highlighted the importance of the drug NCX4040 and its impact on ovarian cancer cell lines. It was impressing to see that the gene expression analysis was done by microarray analysis and further confirmed by RT-PCR method. However, it would be better to have following clarification for further benefits the readers in the field.
1. Why the two cell lines (OVCAR-8 and NCI/ADR-RES) were chosen for the study?
2. How does the concentration of the drug 5 micro-molar was chosen?
3. Some of the tables/ Example table 5 does not have the data for 24 hours time point. Please explain.
4.Figure 4, for the co-expression pattern, it is hard to interpret the results. does any of the gene pattern significantly changed by time point?
5. Any cytotoxic/proliferation experiments were conducted before for this drug ? Please advice.